# An Automatic Velocity Analysis Method for Seismic Data-Containing Multiples

Junming Zhang [1,2], Deli Wang [1,2], Bin Hu [1,2,*] and Xiangbo Gong [1,2]

1 Key Laboratory of Geoghysical Exploration Equipment, Ministry of Education, Jilin University, Changchun 130026, China
2 College of Geo-Exploration Science and Technology, Jilin University, Changchun 130026, China
* Correspondence: binhu@jlu.edu.cn

**Abstract:** Normal moveout (NMO)-based velocity analysis can provide macro velocity models for prestack data processing and seismic attribute inversion. Datasets with an increasing size require conventional velocity analysis to be transformed to a more automatic mode. The sensitivity to multiple reflections limits the wide application of automatic velocity analysis. Thus, we propose an automatic velocity analysis method for seismic data-containing multiples to overcome the limit of multiple interference. The core idea of the proposed algorithm is to utilize a multi-attribute analysis system to transform the multiple attenuation problem to a multiple identification problem. To solve the identification problem, we introduce the local similarity to attribute the predicted multiples and build a quantitative attribute called multiple similarity. Considering robustness and accuracy, we select two supplementary attributes based on velocity and amplitude difference, i.e., velocity variation with depth and amplitude level. Then we utilize the technique for order preference by similarity to ideal solution (TOPSIS) to balance weights for different attributes in automatic velocity analysis. An RGB system is adopted for multi-attributes fusion in velocity spectra for visualization and quality control. Using both synthetic and field examples to evaluate the effectiveness of the proposed method for data-containing multiples, the results demonstrate the excellent performance in the accuracy of the extracted velocity model.

**Keywords:** automatic velocity analysis; multiple independent; local similarity; multi-attribute analysis



## 1. Introduction

Velocity is a critical parameter in both seismic data processing and imaging [1]. Methods have been proposed for velocity model-building such as full waveform inversion [2], joint-migration inversion [3], travel-time tomography [4], normal moveout based velocity analysis [5], and even machine learning [6]. Within these methods, normal moveout-based velocity analyses (hereinafter referred to as velocity analysis) are widely used in industry [7] due to their easy implementation, and no need for a prior velocity model. According to Wilson [8], there are two kinds of velocity analysis: one is to build velocity models for stacking the data for zero offset processing which can increase the signal-to-noise ratio of the stacked dataset [9,10]. The other is to obtain the moveout parameter model that can be used either to analyze the kinematic characteristics of the subsurface medium or as the initial model contributor for subsequent processes, such as travel time tomography, full waveform inversion [11], and time migration [12,13]. These applications make velocity analysis an indispensable step in seismic data processing [14]. Traditional velocity analysis requires processors to manually pick peaks in the velocity spectra. As shown in Figure 1a,b, the peak in the velocity spectra corresponds to the events in the seismic data. The coordinates of the peaks in the velocity spectra correspond to the kinematics (travel time and curvature) of the seismic wavefield, which means that the subsurface information can be obtained by picking manually the coordinate of the peaks in the velocity spectra. However,

with the application of high-precision and even three-dimensional acquisition methods, in the face of massive data, manual picking in the velocity analysis can become a huge burden [15], which leads to that many automatic velocity analysis methods [16–18].

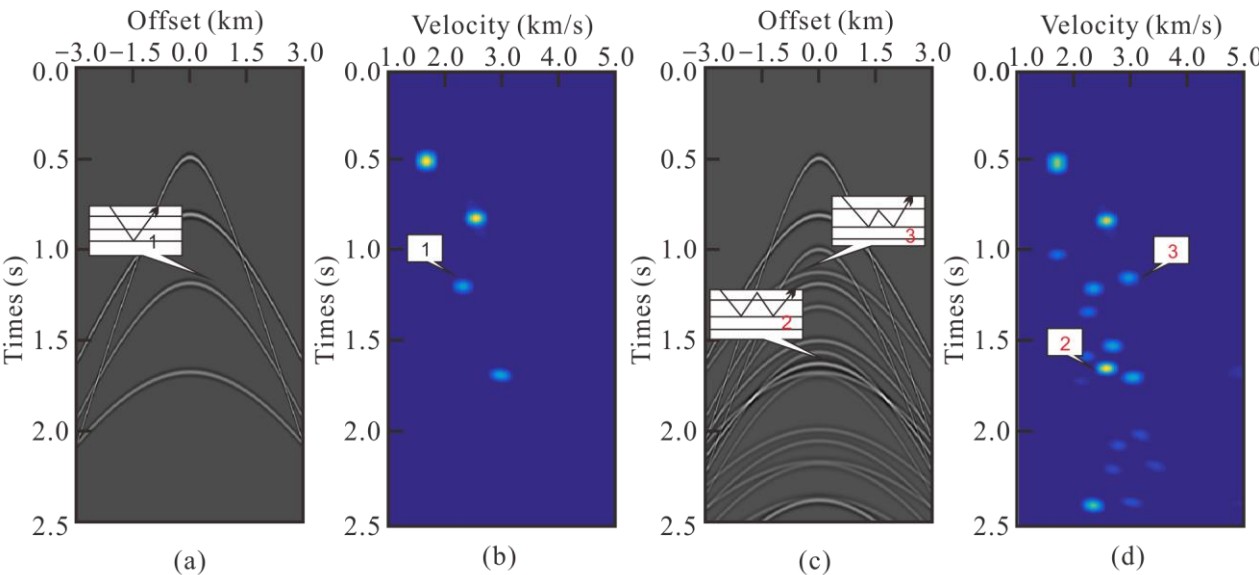

**Figure 1.** Comparison between seismic data without and with multiples and their velocity spectra, There is a corresponding relationship between event and peak marked with the same number in the figure, where the number 1 is primary, number 2 is surface-related multiple, and number is internal multiple: (**a**) seismic data containing only primaries. (**b**) Velocity spectra corresponding to data (**a**). (**c**) Seismic data containing primaries and multiples. (**d**) Velocity spectra corresponding to data (**c**).

There are generally three ways of automatic velocity analysis. The first one is improving the quality of velocity spectra, which modifies the conventional semblance calculation [9] to increase the resolution of velocity spectra [19–21] and the adaptability of complex conditions [5,18]. The second one is optimizing the automatic picking algorithm and utilizing Monte Carlo [17], greedy algorithms [1], clustering analysis [22], simulated annealing [23] and other optimization algorithms to obtain high-precision automatic picking results. The third one is differential semblance velocity analysis (DSVA) [24,25], whose core conception is update migration velocity and flattening images automatically based on the residual moveout of neighboring traces in the image domain [26,27]. Because the accuracy of migration velocity depends on the approximation of the propagation theory to the seismic wavefield propagated in the underground medium, wave equation migration velocity analysis (WEMVA) has become popular [28,29].

Most automatic velocity analysis methods tend to chase higher peaks [30], which are sensitive to coherent noise, especially multiples [5,23,26]. Because the sea surface and the seabed are strong reflective interfaces, multiples cannot be avoided in marine seismic data. The kinematic characteristics of multiples are similar to those of primaries, so they appear as false peaks (Figure 1c,d) in the velocity spectra, which hinder the automatic velocity analysis methods from picking the true velocities. Compared with primaries, multiples reflect more than once between the sea surface and underground interface, and have more complex propagation paths. In Figure 1, two types of multiples are marked. The number two is marked as free surface multiples, and the number two is marked as an internal multiple. The existence of multiples will cause problems in automatic velocity analysis; for example, when using the automatic velocity analysis method to pick the false peaks generated by multiples, the obtained velocity model may be significantly different from the real formation velocity. Furthermore, when this unreliable model is used as the initial model of subsequent inversion such as FWI, it will not meet the accuracy requirements of the inversion [11], which will lead to inversion being trapped in a local non-informative

minimum. So, the traditional seismic data processing requires multiple attenuations before velocity analysis. However, with the development of cognition and technology, in some industrial processes, multiples are regarded as effective signals to improve the imaging accuracy of complex structures, for example, imaging with multiples to reduce the relative amplitudes of the cross-talk noise [31] and enhance the amplitudes in the presence of strong scattering layers [32]. This means that velocity analysis of data containing multiples may become more common [33].

The existing automatic velocity analysis methods rely on two common strategies for dealing with multiples [12,34]. The first category is adopting a muting function in the velocity spectra based on the velocity difference, which assumes that multiples are low-velocity noise [35]. The muting-based strategy can be easily implemented [36], but not applicable to complex situations; for example, it can lead to small differences between primaries and multiples by the occurrence of shallow, high-velocity layers [37], which means the muting-based strategy is difficult to use to completely separate primaries and multiples [38]. The other method is to attenuate multiples before velocity analysis, through appropriate de-multiplying methods to obtain multiple-free input data. This approach can handle complex geological conditions and, thus has wider applicability [35,38]. The picking accuracy of the attenuation strategy depends on the success of multiple attenuation. Wave-equation-based multiple elimination (WEBME) is the most mainstream multiple attenuation method in the industry [39,40], which can be divided into two steps: multiple prediction and matching subtraction. Multiple prediction approximates the travel time of the different types of multiples, i.e., surface-related multiples [41], internal multiples [42], etc. Matching subtraction attenuates the prediction difference in amplitude and phase [43–45]. This means that obtaining multiple-free data is not a simple process, requiring a lot of manual intervention and computational costs [46]. Since currently published methods have some limitations in picking accuracy and automation, and multiples are unavoidable in marine exploration, studying an automatic velocity analysis method for seismic data containing multiples is meaningful.

In order to get rid of the requirement of automatic velocity analysis in primary-only input, we propose an automatic velocity analysis method for seismic data containing multiples. Firstly, we introduce the theory of traditional automatic velocity analysis and then describe how to reduce the sensitivity of automatic velocity analysis to multiples. The multi-attribute analysis system we built consists of multiple similarity, and velocity variation with depth and amplitude level to distinguish between primaries and multiples. Then, we adopt synthetic and field data examples to verify and evaluate the proposed method. Finally, we discuss the advantages and disadvantages of the proposed method, application recommendations and potential research directions.

## 2. Materials and Methods

### 2.1. Common Automatic Velocity Analysis

During automatic velocity analysis, the data-set ordered in shot gathers should be rearranged to common midpoint (CMP) gathers, and the velocity spectra corresponding to the data can be obtained through [9]

$$\mathbf{S}(v,t) = \frac{\sum\limits_{j=-t_{\text{win}}}^{t_{\text{win}}} \left( \sum\limits_{i=0}^{N-1} a[k+j, x_i] \right)^2}{N \sum\limits_{j=-t_{\text{win}}}^{t_{\text{win}}} \sum\limits_{i=0}^{N-1} a[k(k+j, x_i)]^2} \tag{1}$$

where $t$ and $k$ is time index, $k = \sqrt{t^2 + \dfrac{x_i^2}{v^2}}$, $v$ is the scanning velocity, $x_i$ is the offset of trace $i$, $a(k+j, x_i)$ is the amplitude of the data, $t_{\text{win}}$ is the length of the time window, and $N$ is the number of traces.

Then, the velocity curve $L_1[v(t)]$ can be automatically obtained by picking up the peaks in the velocity spectra $\mathbf{s}(t, v(t))$, for example, picking the peaks corresponding to the maximum of the variational integral [17]

$$L_1[v(t)] = \int_{t_{\min}}^{t_{\max}} \mathbf{s}(t, v(t)) dt, \tag{2}$$

where $t_{\max}$, $t_{\min}$ are the upper and lower boundaries of the time window, $v(t)$ is the velocity corresponding to time sample $t$. As mentioned above, in the multiple attenuation process, it is necessary to make the predicted multiples match the actual situation in amplitude and phase to ensure that the multiple can be completely attenuated. Otherwise, we will need strict requirements in the selection of parameters and bounds to avoid the multiples, which is a non-adaptive and time-consuming task. Considering the calculation efficiency, we propose a method that can adaptively use the rough predicted multiples, which are obtained only by simple parameter adjustment and without matching subtraction, to perform automatic velocity analysis on seismic data containing multiples. To sum up, the key to this problem is how to eliminate the influence of multiple peaks on the automatic method by using the differential predicted multiples.

### 2.2. Multiple Independent Automatic Velocity Analysis

In order to eliminate the influence of multiples, we propose a multiple independent automatic velocity analysis method including the following steps:

#### 2.2.1. Peak Picking

We first need to transform the data from shot gathers to velocity spectra. The original data containing multiples and the predicted multiples generated by modularization can be calculated by Equation (1) to obtain the corresponding velocity spectra. We denote the original data velocity spectra as $\mathbf{D}$ and the predicted multiple velocity spectra as $\mathbf{M}$. As mentioned above, when the seismic data contain multiples, there will be many false peaks in the velocity spectra. Thus, it is unreasonable to pick higher peaks at each time step. In this paper, we choose the dichotomy method, pick all the peaks first, and then distinguish the peaks of primaries and multiples. Therefore, in this method we use a robust local maxima finder to find peaks of the velocity spectra [47]

$$\varphi = \mathrm{findpeaks}(\mathbf{D}), \tag{3}$$

where $\varphi = \{(v, t) | (v_1, t_1), \ldots, (v_i, t_i), \ldots, (v_z, t_z)\}$, $i = 1, 2, \cdots, z$ and $z$ is the number of peaks. The coordinate of each peak in the velocity spectra corresponds to its time sample indices and velocity, in which $v, t$ expresses the velocity and time coordinates of peaks, respectively. This process can be easily implemented with the help of the FastPeakFind function in Matlab.

#### 2.2.2. Attribution of Predicted Multiples

We found that, although the rough predicted multiple was obtained without fine parameter adjustment, the actual multiples in the original data are still similar to the predicted multiples, especially in the kinematic characteristics (travel time and curvature). The similarity is much greater than the similarity between the primary and the predicted multiples. In this case, the problem of how to distinguish primaries and multiples in the original data can be solved by measuring the similarity with the predicted multiples. To achieve this goal, this paper proposes a method based on local similarity to attribute the predicted multiples. We assume that the original data velocity spectra is composed of primaries, multiples and noise:

$$\mathbf{D} = \mathbf{D}_p + \mathbf{D}_m + \mathbf{n}_1, \tag{4}$$

where $\mathbf{D}_p$ are primaries, $\mathbf{D}_m$ are multiples, and $\mathbf{n}_1$ is noise. Similarly, the predicted multiple velocity spectra can be written as:

$$\mathbf{M} = \mathbf{M_m} + \mathbf{n}_2, \tag{5}$$

where $\mathbf{M}_m$ are the rough predicted multiples, $\mathbf{n}_2$ is noise. Figure 2a,b show the velocity spectra of original data and the velocity spectra of the predicted multiple, where the red arrows indicate the primary locations and the purple arrows indicate the multiple location. We can see intuitively that the predicted multiples and the actual multiples match in number and locations, while the primaries do not. The reason for the above phenomenon is that the kinematic characteristics of the predicted multiples are similar to the actual multiples and orthogonal to the primaries. This difference is magnified in the velocity spectra.

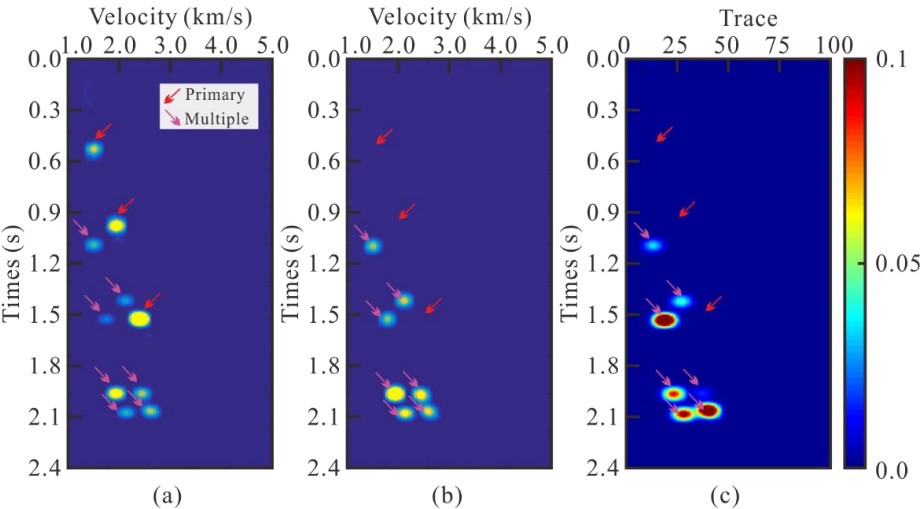

**Figure 2.** A demonstration of the similarity between the original data velocity spectra and the predicted multiple velocity spectra: (**a**) the velocity spectra of original data. (**b**) The velocity spectra of predicted multiples. (**c**) Local similarity spectra.

In order to further analyze the similarity between the original data velocity spectra and predicted multiple velocity spectra, we introduce the local similarity for quantitative measurement [48]. Seismic data evaluated by local similarity are neither instantaneous nor global. Each datum has its corresponding local space; it can be summarized as a certain range, including an element and its adjacent elements [49]. The local similarity is given by:

$$\mathbf{s}_j = \sqrt{\mathbf{c}_{1j} \circ \mathbf{c}_{2j}}, \tag{6}$$

$$\mathbf{c}_{1j} = \delta[\lambda_1^2 \mathbf{I} + \delta^T \left( \mathbf{A}_j{}^T \mathbf{A}_j - \lambda_1^2 \mathbf{I} \right) \delta]^{-1} \delta^T \mathbf{A}_j{}^T \mathbf{b}_j, \tag{7}$$

$$\mathbf{c}_{2j} = \delta[\lambda_2^2 \mathbf{I} + \delta^T \left( \mathbf{B}_j{}^T \mathbf{B}_j - \lambda_2^2 \mathbf{I} \right) \delta]^{-1} \delta^T \mathbf{B}_j{}^T \mathbf{a}_j, \tag{8}$$

where vector $\mathbf{s}_j$ is local similarity between vectors $\mathbf{a}_j$ and $\mathbf{b}_j$. $\mathbf{a}_j$ and $\mathbf{b}_j$ are the elements of the input data at trace index $j$ respectively, $j = 1, 2, \cdots, N$, and $N$ is the number of trace. $\mathbf{c}_{1j} \circ \mathbf{c}_{2j}$ denotes the Hadamard product between $\mathbf{c}_{1j}$ and $\mathbf{c}_{2j}$. $\mathbf{A}_j$ and $\mathbf{B}_j$ are diagonal operators composed from the elements of $\mathbf{a}_j$ and $\mathbf{b}_j$, that is, $\mathbf{A}_j = \mathrm{diag}(\mathbf{a}_j)$, and $\mathbf{B}_j = \mathrm{diag}(\mathbf{b}_j)$, $\delta$ is a triangle smoothing operator aiming to increase the smoothness in iterative optimization schemes [50,51], $\mathbf{I}$ is the identity matrix. $\lambda_1$ and $\lambda_2$ are two parameters controlling the physical dimensionality and enabling fast convergence when inversion is implemented iteratively. For a 2D case, the local similarity is calculated trace by trace, and finally arranged into a matrix $\mathbf{S}$ (Local similarity spectra), which has the same dimension as the input data, that is $\mathbf{S} = [\mathbf{s}_1, \mathbf{s}_2, \cdots, \mathbf{s}_N]$.

Figure 2c shows the local similarity spectra between the predicted multiple and the original data. We can see that the local similarity highlights the similarity regions, in which high amplitude zones indicate multiples (purple arrows) and low amplitude zones indicate primaries (red arrows). In order to quantitatively show the similarity differences between multiples and primaries, we extracted the local similarity calculation results of all peak locations in Figure 2c, and the results are shown in Figure 3.

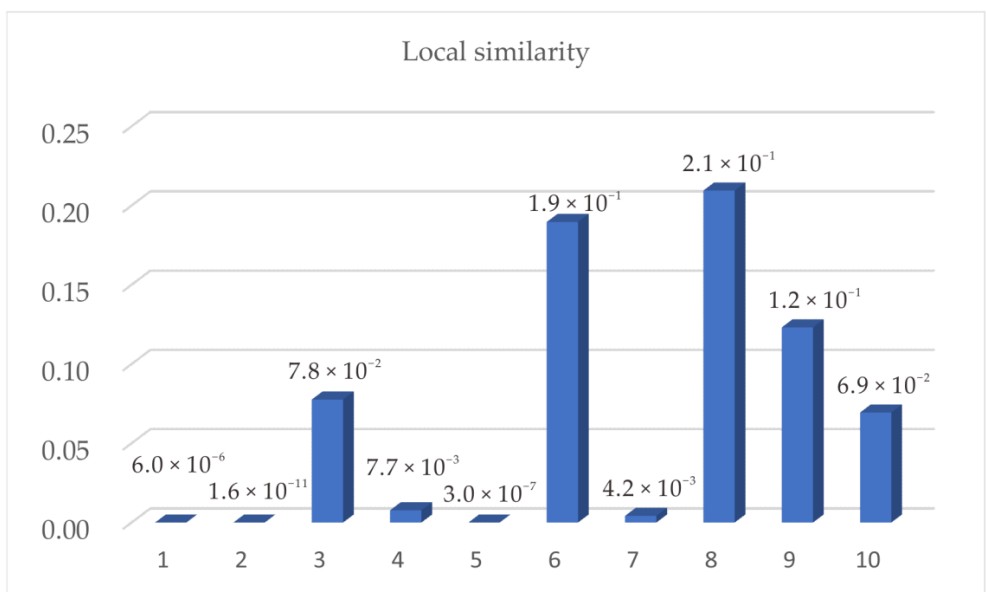

**Figure 3.** Local similarity of the picked peaks.

In Figure 3, the peaks are sorted according to their travel time. Among them, the first, second and fifth peaks are primaries, and the local similarities of these peaks are obviously different from the other peaks. Intuitively, the fourth peak and the seventh peak seem to have a low degree of similarity, but through the quantitative comparison, we can see that there is still an order of magnitude difference between the two peaks and the corresponding value of the primary peaks. In conclusion, even with the smallest multiple and largest primary, there is a difference of more than two orders of magnitude between them.

We calculated the maximum, minimum, average and median of their similarity (Table 1). They also prove that the primaries and multiples in the original data differ greatly in their similarity with the predicted multiple:

**Table 1.** Local similarity statistics between primaries and multiples.

|  | Maximum | Minimum | Average | Median |
|---|---|---|---|---|
| Primary | $6.0 \times 10^{-6}$ | $1.6 \times 10^{-11}$ | $2.1 \times 10^{-6}$ | $3.0 \times 10^{-7}$ |
| Multiple | $2.1 \times 10^{-1}$ | $4.2 \times 10^{-3}$ | $9.7 \times 10^{-2}$ | $7.8 \times 10^{-2}$ |

Note that, in principle, the predicted multiples do not include primaries, which means, compared with the primaries, multiples in the original data are much more similar to the predicted multiples. That is

$$\mathbf{L}[\mathbf{D}_m, \mathbf{M}_m] \gg \mathbf{L}[\mathbf{D}_p, \mathbf{M}_m], \tag{9}$$

where $\mathbf{L}[\mathbf{x}, \mathbf{y}]$ denotes the local similarity between data $\mathbf{x}$ and $\mathbf{y}$. To remove the multiple interference, we took this difference and introduced it into the automatic velocity analysis method. According to Equations (4) and (5)

$$\mathbf{L}[\mathbf{D}, \mathbf{M}] = \mathbf{L}[(\mathbf{D}_p + \mathbf{D}_m + \mathbf{n}_1), (\mathbf{M}_m + \mathbf{n}_2)]. \tag{10}$$

We assume that the noise is distributed separately, then

$$\mathbf{L}[\mathbf{D}, \mathbf{M}] = \mathbf{L}\big[(\mathbf{D}_p + \mathbf{D}_m), (\mathbf{M}_m)\big] = \mathbf{L}\big[\mathbf{D}_p, \mathbf{M}_m\big] + \mathbf{L}[\mathbf{D}_m, \mathbf{M}_m]. \tag{11}$$

According to Equation (9), Equation (11) can be further expressed as

$$\mathbf{L}[\mathbf{D}, \mathbf{M}] = \mathbf{L}\big[\mathbf{D}_p, \mathbf{M}_m\big] + \mathbf{L}\big[\mathbf{D}_m, \mathbf{M}_m\big] \approx \mathbf{L}\big[\mathbf{D}_m, \mathbf{M}_m\big], \tag{12}$$

which means that the local similarity between the original data and the predicted multiples can be used as a reference to distinguish primaries without suppressing the multiples. Therefore, for any original data velocity spectra and predicted multiple velocity spectra, we can calculate their local similarity spectra to distinguish primaries and multiples; this equation is expressed as

$$\mathbf{S} = \mathbf{L}[\mathbf{D}, \mathbf{M}] \tag{13}$$

where $\mathbf{S}$ is the local similarity spectra between the predicted multiple and the original data. Considering the difference between the values of the primary and multiple in local similarity spectra is very large, we attribute them in the logarithmic form.

$$MS_i = \lg(\mathbf{S}(t_i, v_i)), \tag{14}$$

$MS_i$ is the multiple similarity (MS) attribute of the $i$ th peak. $\lg(\mathbf{S}(v_i, t_i)) = \log_{10}(\mathbf{S}(v_i, t_i))$. Through Equation (12), we can quantitatively obtain the similarity between each peak and the predicted multiple.

### 2.2.3. Peaks Identification of Primary Reflection Based on the Multi-Attribute Analysis

The proposed multiple similarity provides a theoretical basis to select the peaks corresponding to the primary reflection from all the peaks. To improve the identification accuracy and robustness of the primary reflection peaks, we propose two supplementary attributes based on other multiple identification principles (velocity and amplitude), and introduce the multi-attribute analysis system [52] as a comprehensive evaluation tool to integrate different picking principles.

The multi-attribute analysis system can integrate and sort the criteria values of different strategies to achieve the selection of the suitable strategy under various qualitative/quantitative criteria [53]. This system mainly includes three steps: attribute definition and selection, determining attribute weights, and comprehensive evaluation [54]. To optimize the multi-attribute structure and reduce information redundancy as well as interference, it is necessary to ensure that all attributes have low correlation and relative independence in the multi-attribute analysis [55].

In this paper, we regard peaks of reflection wavefield (both primary and multiple) as different strategies of peaks, picking for the primary reflection and establishing a corresponding multi-attribute analysis system to decision-making in velocity analysis. The attributes used to distinguish between primary and multiple reflections in the velocity analysis mainly include velocity constraints [34], amplitude [56], bootstrapped differential semblance (BDS) [57] and detecting residual normal moveout [58]. Considering the calculation cost and the difficulty of adaptation, we selected the velocity and amplitude as supplementary attributes, and the three selected attributes are independent of each other, which ensures the stability of the multi-attribute analysis structure. The acquisition methods of the two supplementary attributes are as follows.

(1) Velocity variation with depth (VVD):

This attribute is mainly used to distinguish primaries and multiples through their velocity difference. The principle is that in general, the formation velocity increases with the increase in depth, while the propagation depth of multiples is less than that of primaries,

which leads to the velocity of multiples being less than primaries. For any peak $\varphi_i$, the velocity variation with depth, $VVD_i$, can be expressed as

$$VVD_i = \frac{v_i}{V_{ref}(t_i)}, \tag{15}$$

where $v_i$, $t_i$ express are the velocity and time coordinates of peaks respectively, and $V_{ref}(t_i)$ represents the values corresponding to the velocity reference curve at the $t_i$ time sample indices. Furthermore, the velocity reference curve takes the coordinates of the velocity maximum peaks in different time windows as a reference and is obtained by interpolation along the time direction.

(2) Amplitude level (AL):

Amplitude difference is a classic attribute that mainstream automatic velocity analysis methods tend to use, as they pick the peaks with a higher amplitude [38]. Here, for peak $\varphi_i$, its amplitude level attribute $AL_i$ is given by

$$AL_i = \frac{\lg\mathbf{D}(v_i, t_i)}{\lg\max(W_i)}, \tag{16}$$

where $\lg(\mathbf{D}(v_i, t_i)) = \log_{10}(\mathbf{D}(v_i, t_i))$, $W_i = \{\mathbf{D}(v, t) | t \in [t_i - t_{win}, t_i + t_{win}]\}$, and $t_{win}$ represent the half width of the time window. Considering the thickness of the formation, the time window is set at 100 ms [59].

Note that the above three attributes have different units of measurement, value levels and physical meanings, which need to be regularized before determining attribute weights [60]. In multi-attribute analysis, different attributes can be divided into several categories, corresponding to different regularization equations. Among the three attributes applied by this method, VVD and AL are considered benefit attributes and MS is considered a cost attribute [61]. The regularization equations corresponding to the three attributes is as follows:

$$E_{Li} = \frac{SM_i - SM_{\min}}{SM_{\max} - SM_{\min}}, \tag{17}$$

$$E_{Vi} = \frac{VVD_{\max} - VVD_i}{VVD_{\max} - VVD_{\min}}, \tag{18}$$

$$E_{Ai} = \frac{AL_{\max} - AL_i}{AL_{\max} - AL_{\min}}, \tag{19}$$

where the values in the subscript indicate the maximum and minimum attribute values of all peaks, $E_{Li}$ indicates the value of the $i$ th peak under the condition of MS evaluation. Similarly, $E_{Vi}$ and $E_{Ai}$ indicate the value of the $i$ th peak under the condition of VVD and AL evaluation.

Through the above steps, we can regularize three different attributes. Then, this paper applies the technique for order preference by similarity to ideal solution (TOPSIS) method [62] to integrate the three attributes to automatically obtain the velocity curve. TOPSIS is convenient for application and calculation, and can output a quantitative evaluation [63]. The main steps are listed as follows [64]:

(1) Determine the optimal peak $\varphi_o$ and the worst peak $\varphi_w$;
(2) Obtain the relative proximity $f(i)$ between each peak and the optimal peak;
(3) Determine the dividing point of primary and multiple reflections through the inflection point of evaluation results [65].

In this method, there are $z$ peaks to be evaluated and each peak has three evaluation criteria, whereby the smaller the value of each attribute after regularization, the closer to be primary the event is. Therefore, the optimal peaks $\varphi_o$ and the worst peaks $\varphi_w$ can be defined as

$$\varphi_o = (\min\{E_{L1}, E_{L2}, \ldots E_{Lz}\}, \min\{E_{V1}, E_{V2}, \cdots E_{Vz}\}, \min\{E_{A1}, E_{A2}, \ldots E_{Az}\}), \tag{20}$$

$$\varphi_w = (\max\{E_{L1}, E_{L2}, \ldots, E_{Lz}\}, \max\{E_{V1}, E_{V2}, \cdots, E_{Vz}\}, \max\{E_{A1}, E_{A2}, \cdots, E_{Az}\}). \quad (21)$$

$f(i)$ denotes the calculation of the relative proximity of the $i$ th peak to the optimal peak. It is calculated for each alternative and is defined as

$$f(i) = \frac{\beta_i}{\alpha_i + \beta_i} \quad i = 1, 2, \ldots, z \quad (22)$$

where $\alpha_i$ represents the distance from the $i$ th peak to the optimal peak $\varphi_o$, and $\beta_i$ represents the distance from the $i$ th peak to the worst peak $\varphi_w$. We rank the peaks in descending order according to their corresponding $f(i)$. Through the equation, the primary peaks are those which are closest to the optimal peak and furthest from the worst peak [66]. $f(i)$ is always between zero and one and one peak could be primary when it is closer to one [67]. $\alpha_i$ and $\beta_i$ can be calculated by the following equations:

$$\alpha_i = \sqrt{\omega_L (E_{Lo} - E_{Li})^2 + \omega_V (E_{Vo} - E_{Vi})^2 + \omega_A (E_{Ao} - E_{Ai})^2}, \quad (23)$$

$$\beta_i = \sqrt{\omega_L (E_{Lw} - E_{Li})^2 + \omega_V (E_{Vw} - E_{Vi})^2 + \omega_A (E_{Aw} - E_{Ai})^2}, \quad (24)$$

where $E_{Lo}$, $E_{Vo}$, $E_{Ao}$ are the three attribute values of the optimal peak $\varphi_o$, $E_{Lw}$, $E_{Vw}$, $E_{Aw}$ are the three attribute values of the worst peak $\varphi_w$, $\omega$ is the weight of the attribute. In this paper, we used the analytical hierarchy process (AHP) [68] method to determine the weight of each attribute. The specific choice is $\omega_L = 0.6$, $\omega_V = 0.2$, $\omega_A = 0.2$, the detailed calculation process is highlighted in the Appendix A.

In addition, for conventional velocity analysis, quality control is difficult; once an error occurs, the processor needs to check all the peaks, which is a large burden. Therefore, this method visualizes the automatic velocity analysis process by using an RGB system [69] during processing. The three attributes of the peak are captured into a two-dimensional chart by color. Different colors in the RGB spectra represent different attributes. In this way, the difference between multiples and primaries can be directly displayed by color, so that the processor can focus on key peaks, which reduces the cost and difficulty of manual correction. The process of the proposed method shown in Algorithm 1.

---

**Algorithm 1:** Multiple Independent Automatic Velocity Analysis Algorithm

---

(1) **Input:** Original data containing multiples **d**, Predicted multiple generated by modularization **m**

(2) Calculate the velocity spectra for: **D, M**

(3) Peak picking in velocity spectra **D**: $\varphi \leftarrow \text{findpeaks}(\mathbf{D})$

$\varphi = \{(v, t) | (v_1, t_1), \ldots, (v_i, t_i), \ldots, (v_z, t_z)\}$

(4) Attribute predicted multiples: $\mathbf{S} = \mathbf{L}[\mathbf{D}, \mathbf{M}]$

(5) $i \leftarrow 1$

(6) **Main cycle**:

(9)         Acquire MS attribute: $MS_i = \lg(\mathbf{S}(t_i, v_i))$

(10)         Acquire VVD attribute: $VVD_i = \frac{v_i}{V_{ref}(t_i)}$

(11)         Acquire AL attribute: $AL_i = \frac{\lg s_d(v_i, t_i)}{\lg \max(W_i)}$

(12)             $i \leftarrow i + 1$

(13)     **Until**   $i = z$

(14)     Regularization: $E_{Li} = \frac{SM_i - SM_{\min}}{SM_{\max} - SM_{\min}}$, $E_{Vi} = \frac{VVD_{\max} - VVD_i}{VVD_{\max} - VVD_{\min}}$, $E_{Ai} = \frac{AL_{\max} - AL_i}{AL_{\max} - AL_{\min}}$

(15) Determine primary based on Multi-attribute Analysis theory:

Calculate relative proximity: $f(i) = \frac{\beta_i}{\alpha_i + \beta_i}$, $i = 1, 2, \ldots, z$

Determine the dividing point between primaries and multiples (Inflection point): $f''(i) = 0$

(16) RGB space mapping

(17) **Output**: Velocity curve, RGB velocity spectra

---

### 3. Results

A synthetic example was first used to test the proposed method. In order to increase the variety and complexity of multiples, we used a layered model containing a velocity reversal layer as shown in Figure 4a. The CMP gather is shown in Figure 4b, and Figure 4c is the corresponding velocity spectra. In the CMP gather, we can see that primaries and multiples share the same kinematic characteristics and strong amplitudes, making it difficult to distinguish primaries, especially for 2.5–4.5 s. In the velocity spectra shown in Figure 4c, due to the influence of the low-velocity layer (1.5–2.25 km), the velocities of multiples and primaries are close to each other. Thus, false peaks and primary peaks have similar velocities in the range of 2.5–4.5 s. These false peaks will largely interfere with automatic methods of picking velocities.

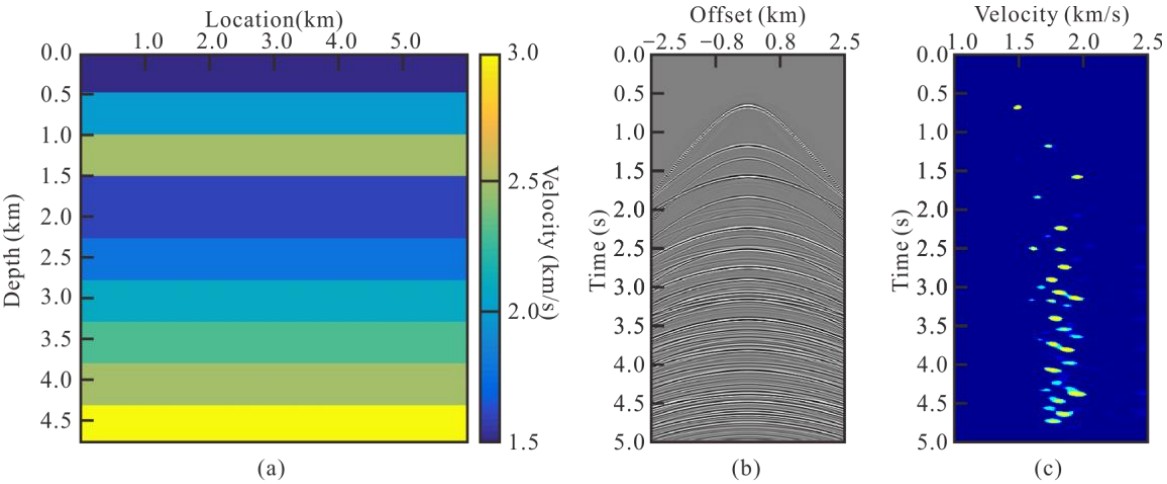

**Figure 4.** Display of synthetic data: (**a**)Velocity model containing a velocity reversal, (**b**) CMP record of original data, (**c**) velocity spectra of original data.

We applied four different automatic velocity analysis strategies in the test of synthetic data; the results are shown in Figure 5. Figure 5a is the result using muting-based strategies (we refer to it as MU), Figure 5b shows the result by radon transform [1] (we refer to it as RT), Figure 5c displays the result based on multiple attenuation strategies (we refer to it as MA), and Figure 5d is the result through the advanced multiple independent method (we refer to it as AD). By comparing the Figure 5 with Figure 4b, we can have a qualitative understanding of the multiple processing capabilities of various strategies. In Figure 5a, The white curve represents the muting range. It can be seen that in the shallow time zone (before 2.5 s), when multiples and primaries are scattered, the muting strategy can filter out the multiples, but in the range of 2.5–4.5 s, since multiple and primary are difficult to distinguish, most multiples are retained to avoid damaging the effective signal. For RT strategy in Figure 5b, the sparse hyperbolic Radon transform provides a sparse domain for velocity analysis; some multiples are attenuated in the transformation process, but there are still considerable residues (especially after 2.5 s). Comparing Figure 5c with Figure 4b, after multiple attenuation, most of the false peaks generated by multiples are attenuated, but there are still small amounts of multiple residuals. Figure 5d is the velocity spectra corresponding to proposed method, in which the peaks of primaries and multiples are displayed in color. In Figure 5, the red curves on each figure are the velocities obtained by the corresponding method. The greater the interference of multiples, the mor fluctuation the velocity curve will have. In order to evaluate the effectiveness of the proposed methods, we have made a quantitative analysis of the four velocity curves.

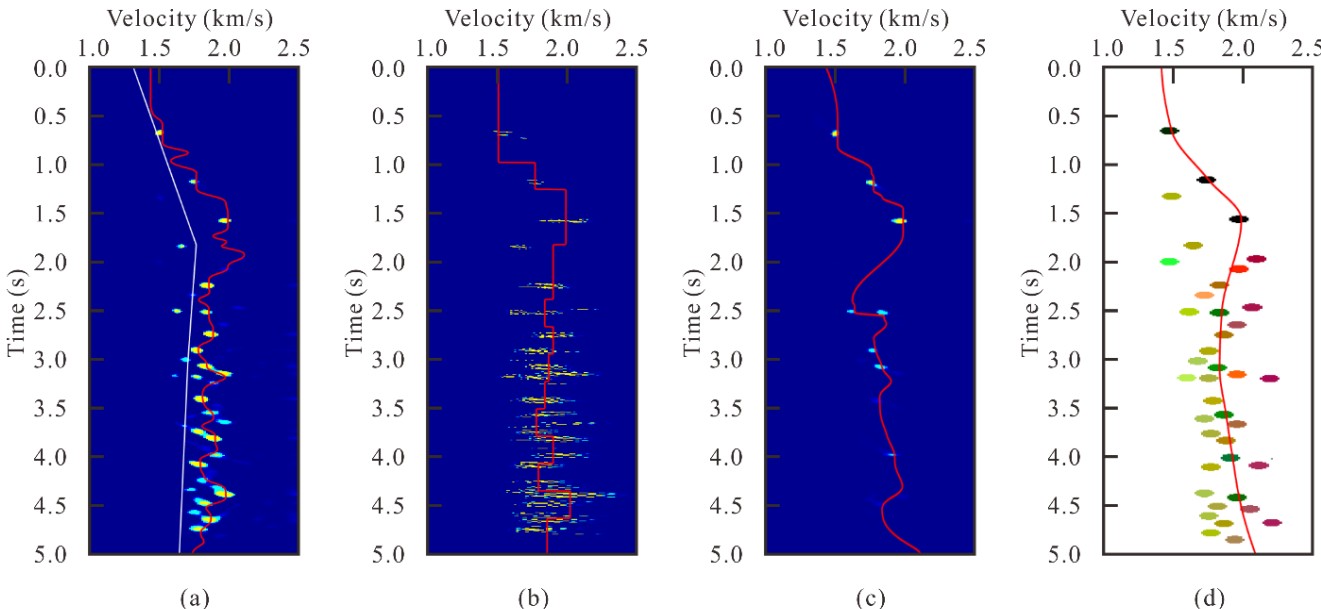

**Figure 5.** Display of synthetic data test results: (**a**) muting-based strategy (MU), (**b**) Radon transform strategy (RT), (**c**) multiple attenuation strategy (MA), (**d**) proposed advanced method (AD).

We manually picked the peaks of primaries referring to the true velocity model, and using the manual picking (refer to it as MP) result as the reference velocity. To test the effectiveness of the proposed method, we compared the velocities obtained by various strategies with the reference curve, and used $\sigma$ to quantitatively measure the difference between the reference and estimated velocities, The results are shown in Figure 6 and Table 2; the equation of $\sigma$ is as follows:

$$\sigma = \frac{\|v_r - v_e\|_2}{\|v_r\|_2}, \tag{25}$$

where $v_r$ is reference velocity and $v_e$ is estimated velocity. In Figure 6, the yellow, black, brown, blue and red lines represent MP, MU, RT, MA and AD velocities, respectively. Corresponding to the quantitative analysis results in Table 2, we can see that the estimated velocity by AD is the closest one to reference and that the red and yellow curves are almost overlapping each other. Correspondingly, Table 2 shows that between the AD velocity and the reference velocity, the calculated metric is $\sigma = 1.3\%$ which is the smallest among the four strategies. The estimated velocity using the MA strategy is suboptimal, for which $\sigma = 5.0\%$. The main reason for the impact on the estimation accuracy can be clearly found in Figures 6 and 5c, That is, the residual multiples at 2.5 s makes the automatic method pick a false peak. Additionally, more false picks are picked by the RT method and MU method, which further reduces the accuracy of their estimated velocities, for which the $\sigma$ are 5.4% and 6.0% respectively. This comparison result shows the dependence of traditional automatic velocity analysis algorithm on multiple attenuations. However, the method proposed in this paper can measure the difference between the actual situation and the predicted multiples through local similarity, effectively improving the tolerance.

In order to test the accuracy of the velocity obtained by proposed method, we respectively used the reference velocity (MP), the velocity obtained by the proposed AD method, and the velocity obtained by the MA strategy, which is suboptimal to perform NMO correction on the original data. For a closeup look, we zoom the data in time regions 1.4–2.4 s and 3.2–4.2 s. The results are shown in Figure 7.

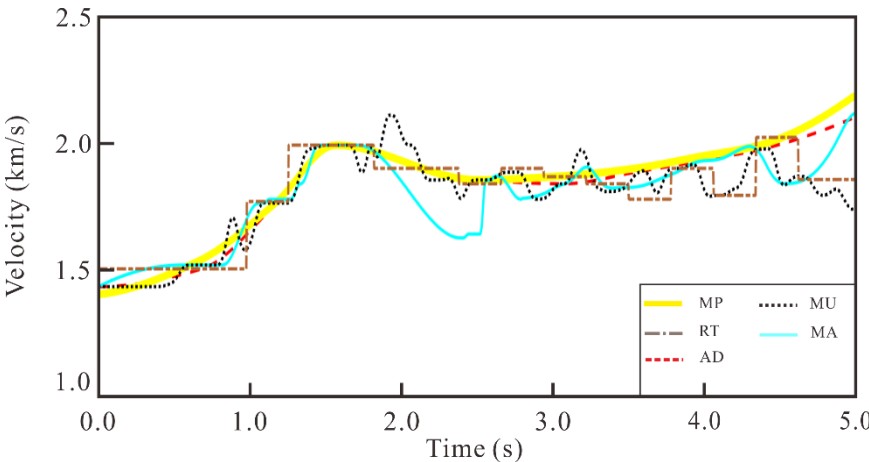

**Figure 6.** Velocity comparison.

**Table 2.** The difference between the reference and estimated velocities.

|  | MB | RT | MS | MI |
|---|---|---|---|---|
| $\sigma$ | 6.0% | 5.4% | 5.0% | 1.3% |

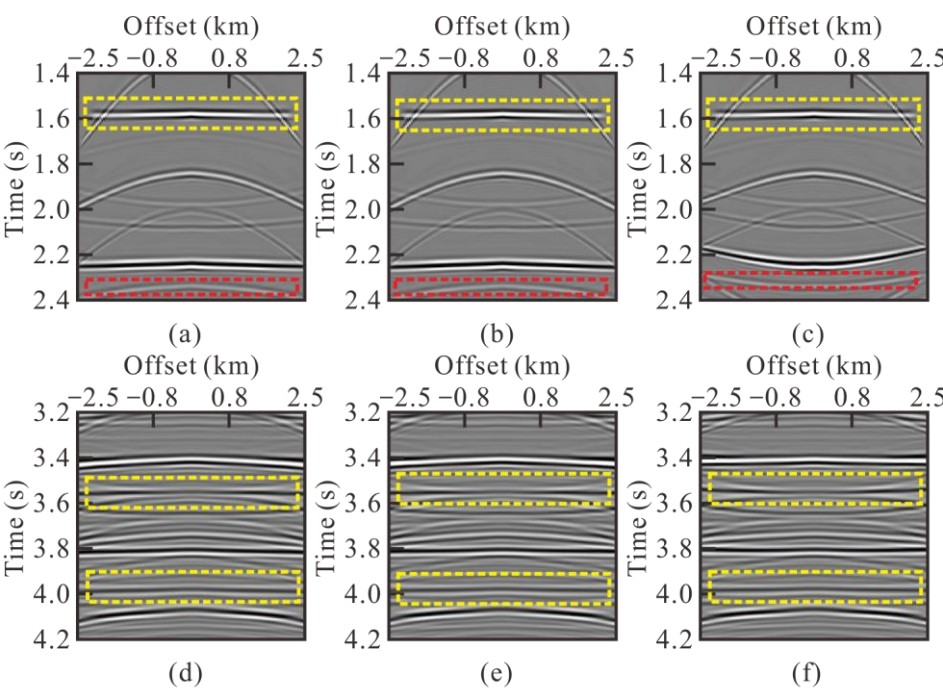

**Figure 7.** NMO comparison results. 1.4–2.4 s: (**a**) NMO results of the MP velocity, (**b**) NMO results of the AD velocity, (**c**) NMO results of the MA velocity. 3.2–4.2 s: (**d**) NMO results of the MP velocity, (**e**) NMO results of the AD velocity, (**f**) NMO results of the MA velocity.

Within 3.2–4.2 s, the primary events were basically flattened (marked by yellow dotted line). However, in the 1.4–2.4 s, due to the influence of the residue false peak, the velocities obtained by MA strategy were lower than the real ones, which led to the overcorrection of the multiple events (marked by red dotted line). So, the results of NMO correction proved the effectiveness of the proposed method for data containing multiples.

In order to further verify the effectiveness of the proposed method, we applied it to field marine data for testing. The proposed method first was applied to a single CMP gather for quantitative analysis, and then we used the proposed method to process all CMP

gathers to obtain the 2D velocity model of field data. For the data, the CMP interval is 25 m, the sampling interval is 4 ms, and maximum time is 2.8 s. Further, in the test on field data, because the real velocity model is unknown, we take the velocity curve picked manually as reference. As the multiple attenuation strategy can achieve better results than the other two strategies, we take the multiple attenuation strategy as the comparison object of the proposed method in the test of field data.

By comparing Figure 8b,c, we can see that after the multiple attenuation processing, the effective signals in the field data have been highlighted, which improved the data quality, but multiple residues can still be observed. The false peaks generated by these multiples will seriously interfere with the automatic velocity analysis. Comparing the velocity curves in Figure 8c,d, we can see that the velocity obtained by the proposed method is closer to the reference velocity. In order to further test the stability of the method, we applied it to the entire data to obtain a velocity model. The velocity model is shown in Figure 9, (a) is the reference model picked manually, (b) is the velocity model obtained by the control group, and (c) is the velocity model obtained by the proposed method. There is an obvious gap in the 1.2–1.6 s area (marked by red dotted line). The reason is that the model obtained by the control group is affected by residual multiples in this area. Meanwhile, in Figure 9b, there is a mixture of high-velocity and low-velocity formation (marked by red arrow), as a consequence of the incorrect picking of low-velocity multiple peaks by the control method.

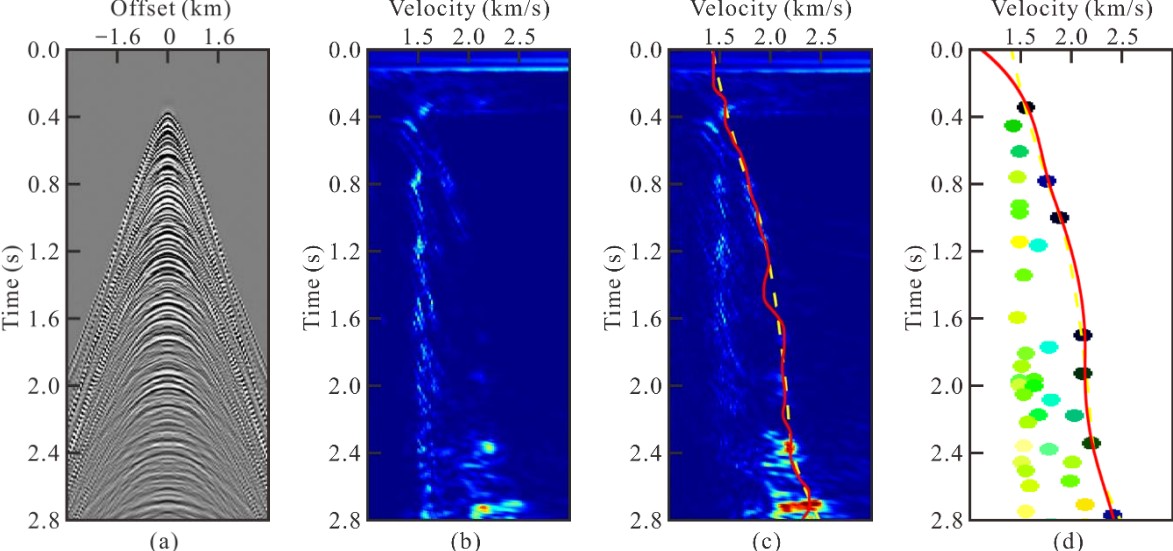

**Figure 8.** Application on a single CMP gather from the field data: (**a**) CMP record of original data, (**b**) velocity spectra of original data, (**c**) velocity spectra and velocity curve of the control group, in which the red solid line is automatically obtained, and the yellow dotted line is the reference velocity, (**d**) velocity spectra and velocity curve obtained by the method proposed in this paper, in which the red solid line is automatically obtained, and the yellow dotted line is the reference velocity.

We respectively applied the three velocity models to the original data to calculate their stacked sections. Figure 10a shows the stack section corrected with reference velocity, (b) is stack section of the control groups and (c) is the stack section with velocity obtained by the proposed method. In comparison, one event exists in Figure 9a,c near 1.6 s (marked by yellow dotted line), but in the stacked section of the control group, this event cannot be observed. The reason is that near 1.6 s, the control method is affected by the residual multiple and obtained the wrong velocity, so that the primary event was covered. In the comparison of three stacked sections, the stacking sections with velocities obtained by the proposed method are basically consistent with the reference-stacked section, which proves the effectiveness of the method for data-containing multiples.

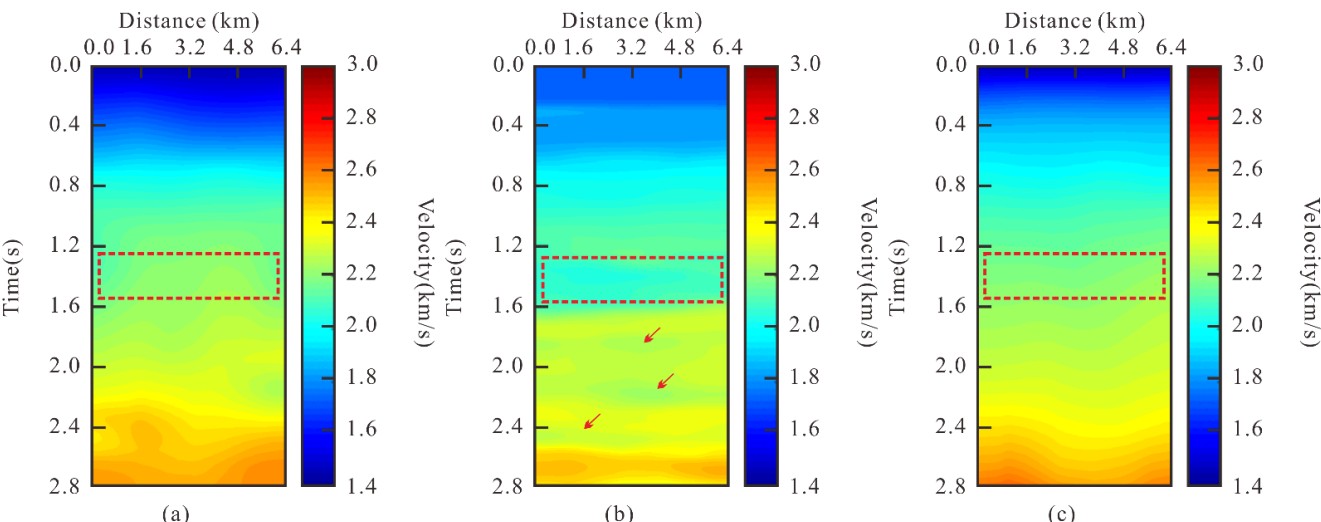

**Figure 9.** Comparison of velocity model: (**a**) reference model, (**b**) velocity model obtained by control group, (**c**) velocity model obtained by proposed method.

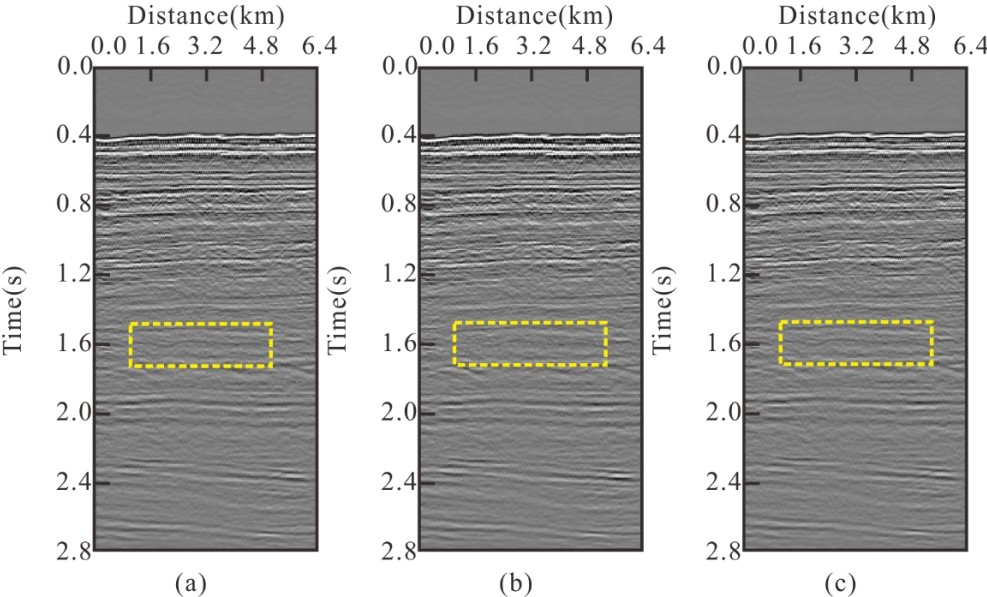

**Figure 10.** Comparison of stacked sections: (**a**) stack result of reference velocity, (**b**) stack result of the velocity obtained by control group, (**c**) stack result of the velocity obtained by proposed method.

## 4. Discussion

NMO-based velocity analysis is an indispensable step for whole seismic exploration, and can provide the initial velocity model for other velocity analysis methods based on the inversion problem [11,13]. The traditional interactive picking method may not meet the processing requirements of massive seismic data. However, the effectiveness of the published automatic methods is degraded by multiple interference. This gap for automatic velocity analysis makes the subject of the proposed method meaningful. Compared with conventional automatic velocity analysis, the proposed method significantly eliminates the influence of multiples and obtain an accurate velocity curve, which has been demonstrated in the synthetic and field examples.

### 4.1. Sensitivity to Multiple

For multiple-free datasets, conventional automatic velocity analysis can obtain a relatively accurate picked velocity in a fully automatic way. The coherent noise in CMPs

greatly affects the peaks picking, leading to a lower picked velocity. To overcome this limit, several methods (e.g., multiple attenuation before velocity analysis [35], muting low-velocity zone in velocity spectra [38], interactive guidance [70], dip filter in CMPs [26]) are proposed, which make the current velocity analysis semi-automatic. In this paper, as to achieve velocity analysis for data-containing multiples in a fully automatic way, we attributed the predicted multiples and combined them with two additional auxiliary attributes through multi-attribute analysis as the novel principle of picking peaks.

### 4.2. Processing Detail

For velocity picking, the accuracy of picked velocity depends on the resolution of the velocity spectra. For improving the resolution, many researchers propose several modified semblances by measuring the difference between the neighbor traces in some domains and building the corresponding time-weighting function, such as AB semblance [18], offset-dependent weighting semblance [5], similarity-weighted semblance [14] and so on. For the proposed method, we adopt the traditional semblance proposed by Taner [9]. The reason for choosing the traditional one is automation and computing efficiency. The resolution of these modified semblance depends on the applicability of the weighting function, which means parameter adjustment and additional calculation costs. However, we still recommend utilizing an advanced semblance calculation in complex situations to improve the stability of automatic velocity analysis, such as data with the amplitude variation with offset (AVO) anomaly.

Among the attributes proposed in this paper, multiple similarity is the most critical. Its basic assumption is that the existing methods can accurately predict the travel time of the multiples. In order to meet the assumption as much as possible, we recommend predicting different types of multiples (e.g., surface-related multiples, internal multiples, water-bottom multiples) by traditional multiple prediction methods in the industry. In this paper, we highlight using the surface-related multiple elimination (SRME) approach [41] to predict surface-related multiples and common-focus-point (CFP) approach [71] to predict internal multiples, etc. The above strategies are obtained by balancing prediction and computational efficiency. In theory, the completeness of predicting multiples determines the accuracy of identifying multiples. We adopted multiple prediction without adjusting parameters to efficiently obtain different types of multiples. Then, we utilized local similarity to ensure the accuracy of the pro-posed automatic velocity analysis under the condition of large prediction error. It is true that when the formation conditions are complex, some special multiples such as diffraction multiples cannot be predicted. At this time, the multiple similarity attributes may not be applicable, but the other two auxiliary attributes can be combined to achieve primary extraction.

### 4.3. Usage Recommendation

For the proposed method, we recommend two types of uses:

(1) Batch processing when the data size is too large. For instance, when processing three-dimensional data, automatic velocity analysis can reduce a lot of burden in the face of massive CMPs. The proposed method is fully automatic and multiple-independent, which is more suitable for adaptive batch processing.

(2) Instead of the traditional semblance spectra, the RGB-based color spectra provide a more intuitive distinction between primaries and multiples. The color of each peak implies rich geophysical information, which can be used as a reference for manual picking. The relationship between seismic wavefield and color is shown in Table 3:

The RGB display allows a more convenient interaction mode in peaks picking. The processor can just focus on the suspicious peaks and obtain a more accurate velocity model with the assistance of color.

**Table 3.** Analysis for the color spectra.

| Type | Features | Color |
|---|---|---|
| Primary | low similarity, high velocity and amplitude in common | ⬤ ⬤ |
| Internal multiple | low amplitude, high similarity. velocity relates to the developed formation. | ⬤ ⬤ ⬤ |
| Surface-related multiple | high similarity, velocity relates to shallow formation, amplitude could be high. | ⬤ ⬤ |
| Suspicious peak | the peak should be focused on | ⬤ ⬤ |

### 4.4. Potential Research Directions

Recently, deep learning has rapidly developed and even surpassed the human level in dealing with some practical problems, such as real-time processing. Automatic seismic inversion and processing with deep learning has gradually become a trend [6,72]. However, the training process requires large numbers of labeled samples [73]. For NMO-based velocity analysis, a standardized processing flow is needed, such as pre-processing, multiple attenuation and manual picking, which are time-consuming and laborious. A large number of high-quality training sample sets can be obtained through the proposed method combined with manual inspection.

Multiples are considered a contribution to imaging, which increases demand for velocity model building of dataset-containing multiples. The proposed method can match this demand and provide a micro-velocity model for follow-up imaging application. In this way, migration velocity analysis based on the wave equation can be extended to data-containing multiples.

### 5. Conclusions

We propose a multiple independent automatic velocity analysis method to solve the problem that automatic picking methods do not work with multiples. The main idea here is the utilization of the concept of multiple similarity as a novel peak-picking principle to break the primary dominant assumption. We combine multiple similarity with classical attributes such as velocity and amplitude as a compound-picking principle and adopt multi-attribute analysis theory to pick adaptively the peaks of the primaries. The proposed method has been applied to synthetic and field data tests, and has achieved better results than the control group, which verifies its effectiveness to data-containing multiples.

Another advantage of the proposed method is the visualization. Different from the traditional semblance spectra, we connect the three proposed attributes to the RGB system and display these peaks by color, leading to one peak that can demonstrate more comprehensive and intuitive geophysical information for processers.

The proposed method has the advantages of automation, efficiency and is easy to implement. Considering the future development trend of seismic exploration, the proposed multiple independent automatic velocity analysis method has practical value and sufficient development potential.

**Author Contributions:** All authors made significant contributions to this paper. J.Z.: algorithm writing, data analysis and original manuscript writing. B.H.: Investigation, development of ideas and review the manuscript. D.W.: Supervision, conceptualization, and field data acquisition. X.G.: Modeling, data test and manuscript editing. All authors have read and agreed to the published version of the manuscript.

**Funding:** This research was supported in part by the National Natural Science Foundation of China (No. 42074151, No. 41374108), and Major Projects of the National Science and Technology of China (Grant No. 2016ZX05026-002-003). This project is supported by special fund of Key Laboratory of Geoghysical Exploration Equipment, Ministry of Education (Jilin University).

**Data Availability Statement:** Not applicable.

**Acknowledgments:** Thanks for the open source program of Matlab.

**Conflicts of Interest:** The authors declare no conflict of interest.

## Appendix A

Analytic hierarchy process (AHP) [74] is used as the common weight determination method in multi-attribute Analysis to determine the weight of each evaluation attribute and the determination process is as shown in the equation

$$\mathbf{O} = \begin{pmatrix} 1 & o_1 & o_2 \\ \dfrac{1}{o_1} & 1 & o_3 \\ \dfrac{1}{o_2} & \dfrac{1}{o_3} & 1 \end{pmatrix}, \tag{A1}$$

where $\mathbf{O}$ represents the decision matrix, $o_1$ is the scale of the first attribute relative to the second attribute, $o_2$ is the scale of the first attribute relative to the third attribute, and $o_3$ is the scale of the second attribute relative to the third attribute.

The specific value of the decision matrix can be obtained by looking up the scale table in the AHP method [74]. AHP method does not need to quantify the importance of each attribute but it only needs to qualitatively determine the importance of each attribute to obtain the scale of each attribute. The VVD and AL attributes are more likely to be affected by complex terrain because they are based on the difference of velocity and amplitude. Because the multiple prediction method is widely used, therefore, it is qualitatively believed that the MS attribute is slightly more important than the others. Moreover, the decision matrix can be converted according to the scale table of AHP method into the form shown in (A2).

$$\mathbf{O} = \begin{pmatrix} 1 & 3 & 3 \\ \dfrac{1}{3} & 1 & 1 \\ \dfrac{1}{3} & 1 & 1 \end{pmatrix}. \tag{A2}$$

After the establishment of the decision matrix, the weight of each attribute in the AHP method can be obtain by normalizing the principal vector (eigen vector) of the matrix [73]. Through this way, it can be found that $\omega_L = 0.6$, $\omega_V = 0.2$, $\omega_A = 0.2$.

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
