# Peer review of "An Automatic Velocity Analysis Method for Seismic Data-Containing Multiples"

_remotesensing, doi:10.3390/rs14215428_

Round 1
Reviewer 1 Report
The manuscript discusses an interesting velocity analysis method for the seismic data containing multiples. It needs reversion to improve its quality.
1. In equations 6-8, C1 and C2 are matrixes, (C1)’C2 is also a matrix, S is a scalar or a matrix?
2. In equations 7-8, give some details about delta.
3. What is s_d in equation 16?
4. It is better to show f(i), and how to use it?
5. What is control group?
6. It is hard to view the velocity spectrum in Figure 5c.
7. It is better to point the overcorrected event in Figure 6c.
8. L12, become->becomes
9. L33, what is velocity analysis for short?
10. L62, “Figure 1. Comparison between primaries and data contains multiples:” -> “Figure 1. Comparison between seismic data without and with multiples and their velocity spectrum:”
11. Delete “:” at the end of Line 94, 117, 120, 135, 218, 231, 243.
12. L119, 2.21, 2.22, and 2.23 should be 2.2.1, 2.2.2, and 2.2.3.
13. L131, “Z is the number of peaks”-> “and Z is the number of peaks”.
14. L218, base->based.
15. L288, finded->found.
16. L370, “Figure8 (c) and Figure8 (d)” should be “Figure7 (c) and Figure7 (d)”.
17. Matrix O has three eigenvectors, it is better to say the eigenvector corresponding to the largest eigenvalue.
Reviewer 2 Report
The manuscript describes an automatic velocity picking tool that avoids multiples by utilizing multiple attributes. The paper is reasonably readable and the proposed approach seems novel and useful. However, I have some concerns with regard to the development and the clarity of some sentences. Let me start with equation 1, the semblance measure. It is described with the symbol “s”, which is utilized in equation 2 as a function of t and v(t). So, for velocity analysis, I assume equation 1 implies the semblance after NMO correction and it is a function of index j. However, above the equation, it is not referred to as a generic semblance measure, but one specifically for the velocity spectrum, so why isn’t “s” in equation 1 a function of velocity. Also, in the definition of symbols, I notice a lot of font changes, like that after equation 3. I am also confused by equations 7 and 8. Since A and B are diagonal matrices, they are equal to their transpose, so why use transpose, which is not even defined. I am not sure why in Figure 1 the vertical axis is set to decrease going up. This has to be mentioned in the caption. In equation 16, why “lg” is used to represent “log”, and it is not even defined. Also, in Figure 7, the attributes in (d) seems to be shifted to later times compared to the location of the strong primaries, which is puzzling. Is there an explanation for that?
I have additional comments, suggestions and corrections in the uploaded annotated version of the paper. I encourage the authors to address them and make the necessary revision to the paper.

Reviewer 3 Report
The abstract does not reflect the content of the own work, rather summarizes the introduction. It needs to be rewritten.
Line 31, replace etc with among others. Than and Line 32 replace “among” with “within”, sounds better.
The introduction refers to several key studies, while other more important ones are missing, such as:
Shen, P., & Symes, W. W. (2008). Automatic velocity analysis via shot profile migration. Geophysics, 73(5), VE49-VE59.
Mulder, W. A., & Ten Kroode, A. P. E. (2002). Automatic velocity analysis by differential semblance optimization. Geophysics, 67(4), 1184-1191.
Weibull, W. W., & Arntsen, B. (2013). Automatic velocity analysis with reverse-time migration. Geophysics, 78(4), S179-S192.
Wiggins, J. W. (1988). Attenuation of complex water-bottom multiples by wave-equation-based prediction and subtraction. Geophysics, 53(12), 1527-1539.
In general I´d say there are some 20-25 important references missing, just in this part.
In the last paragraph of the introduction, when the own purpose of the manuscript should be stated, the authors pre-conclude already their study, prior presentation of their work. This needs to be changed.
The section where the methodology is presented is very long in respect to the entire document and it has parts which stand out of the usual template of Remote sensing, such as since line 297. In the same section the cited references are scarce and many, many key studies are missing such as:
Shevchenko, G., Ustinovichius, L., & Andruškevičius, A. (2008). Multi‐attribute analysis of investments risk alternatives in construction. Technological and economic development of economy, 14(3), 428-443.
Jiménez, A., Ríos-Insua, S., & Mateos, A. (2006). A generic multi-attribute analysis system. Computers & operations research, 33(4), 1081-1101.
Etc
The results section is the highlight of the manuscript, as the own results are nicely presented and explained, while the discussion section is far too short, missing comparisons with similar studies as there is a total of just one (1!!!) single reference. The format of table 2 is not of the correct format and this section should not end with just a table. In other words, lots of needed text is missing here.
Round 2
Reviewer 2 Report
The manuscript has improved and I have no technical concerns. There are minor typos that I point to in the uploaded annotated version pf the paper.

Author Response
Dear reviewer:
Thank you very much for your generosity and kindness. You have provided us with great help in the whole process of revising the manuscript. In the two rounds of revision, you have revised your counterpart's manuscript word by word. We are impressed by your seriousness and responsibility. With your help, the quality of our manuscript has been substantially improved, and it is very different from the previous version in terms of readability and English standardization. Your suggestion this time has been marked in the latest manuscript through the revision mode(see as the attachment). Please let us express our sincere thanks again for your great help in the improvement of our manuscript. I hope that the revised manuscript is now suitable for publication in Remote Sensing.
Sincerely,
Junming Zhang &
Bin Hu
College of Geo Exploration Science and Technology, Jilin University, Changchun 130026
People’s Republic of China
Email address: junming21@mails.jlu.edu.cn
binhu@jlu.edu.cn

Reviewer 3 Report
Thanks for all given improvements. The manuscript should be published in its present form
Author Response
Dear reviewer,
Thank you very much for your approval of our manuscript. In the last revision, your thoughtful suggestions and insights gave us great help. After the revision, the quality of the manuscript has been substantially improved. The readability and richness of the manuscript are very different from the original version. Please let us express our sincere thanks again for your help in the manuscript publishing process.
Sincerely,
Junming Zhang & Bin Hu
College of Geo Exploration Science and Technology, Jilin University, Changchun 130026
People’s Republic of China
Email address: Junming21@mails.jlu.edu.cn binhu@jlu.edu.cn